# Evaluation of a novel point-of-care lateral flow assay screening for *Neisseria gonorrhoeae* infection among pregnant women in Zimbabwe

Kevin Martin[1,2]*, Ethel Dauya[2], Chido Dziva Chikwari[2,3],
Constance R. S. Mackworth-Young[2,4], Victoria Simms[2,3], Tsitsi Bandason[2],
Beauty Makamure[2], Janice Martin[2], Forget Makoga[2], Anna Machiha[5],
Remco P. H. Peters[6,7,8], Laura T. Mazzola[9], Benjamin Blumel[9], Birgitta Gleeson[9],
Michael Marks[1], Katharina Kranzer[1,2,10], Rashida A. Ferrand[1,2], Cecilia Ferreyra[9]

1 Department of Clinical Research, London School of Hygiene & Tropical Medicine, London, United Kingdom, 2 Biomedical Research and Training Institute, Harare, Zimbabwe, 3 Department of Infectious Disease Epidemiology, MRC International Statistics and Epidemiology Group, London School of Hygiene & Tropical Medicine, London, United Kingdom, 4 Department of Global Health and Development, London School of Hygiene & Tropical Medicine, London, United Kingdom, 5 AIDS and TB Unit, Ministry of Health and Child Care, Harare, Zimbabwe, 6 Research Unit, Foundation for Professional Development, East London, South Africa, 7 Department of Medical Microbiology, University of Pretoria, Pretoria, South Africa, 8 Division of Medical Microbiology, University of Cape Town, Cape Town, South Africa, 9 FIND, Geneva, Switzerland, 10 Division of Infectious Diseases and Tropical Medicine, LMU University Hospital, LMU Munich, Munich, Germany

* kevin.martin@lshtm.ac.uk

## Abstract

Affordable, easy-to-use and rapid diagnostics may support a move away from syndromic management for sexually transmitted infections (STIs) in resource-constrained settings. A lateral flow assay for *Neisseria gonorrhoeae* (NG-LFA) has shown high sensitivity and specificity (>90%) in symptomatic individuals. We investigated the performance and acceptability of this assay as a screening tool for NG among pregnant women. This evaluation was embedded within a prospective study evaluating point-of-care STI screening in pregnant women attending antenatal care (ANC) in Harare, Zimbabwe. Participants were included regardless of symptom status, ANC visit number, or gestational age. Nurse-collected vaginal swabs were tested on-site using the NG-LFA and the Xpert CT/NG assay (Xpert) (reference test). The implementation team members (n=4) were interviewed to assess acceptability and usability of NG-LFA. Of 912 participants, 4.8% (44/912) self-reported presence of abnormal vaginal discharge. Xpert NG prevalence was 4.2% (38/912); 81.6% (31/38) of infections were asymptomatic. The sensitivity, specificity, positive predictive value, and negative predictive value (NPV) of the NG-LFA were 65.8% (25/38; 95% CI 48.6%–80.4%), 99.2% (867/874; 95% CI 98.4–99.7%), 78.1% (25/32; 95% CI 60.0-90.7%), and 98.5% (867/880; 95% CI 97.5-99.2%). The NG-LFA was considered easy-to-use and interpret but discordant results led to issues of trust in the NG-LFA results. Among predominantly asymptomatic pregnant women, the NG-LFA had high specificity, but relatively low sensitivity meaning one in three cases of gonorrhoea were

**Data availability statement:** Data and the codebook are available from the London School of Hygiene & Tropical Medicine (LSHTM) Data Compass repository: https://doi.org/10.17037/DATA.00004472.

**Funding:** This work was supported by the Wellcome Trust (225468/Z/22/Z to KM) and via FIND (to KM) by the UK Department of Health and Social Care as part of the Global AMR Innovation Fund (GAMRIF), by the Foreign, Commonwealth and Development Office of the UK government and with support from the Ministry of Foreign Affairs of the Government of the Netherlands. GAMRIF is a One Health UK aid fund that supports research and development around the world to reduce the threat of antimicrobial resistance in humans, animals and the environment for the benefit of people in low- and middle-income countries. The views expressed in this publication are those of the author(s) and not necessarily those of the UK Department of Health and Social Care and they do not necessarily reflect the UK government's official policies. The funders had no role in study design, data collection and analysis, decision to publish, or preparation of the manuscript.

**Competing interests:** LTM is employed as a consultant by FIND. BB, BG and CF are employees of FIND. All other authors declare no competing interests. This does not alter our adherence to PLOS ONE policies on sharing data and materials. The prototype lateral flow assay and reader evaluated in this study were developed by FIND, in collaboration with the contract research organization DCN Dx, to facilitate gonorrhoea/chlamydia testing at the point of care in low- and middle-income countries. FIND aims to transfer the technology under a licensing agreement with Global Access terms to a commercial manufacturer. FIND is a global nonprofit connecting countries and communities, funders, decision makers, health care providers, and developers. FIND will not receive royalties from the sales of the final assay.

not detected. Further studies are warranted to assess the clinical performance and cost-effectiveness of the NG-LFA in other settings and populations.

## Introduction

The World Health Organization (WHO) has recommended syndromic management of sexually transmitted infections (STIs) in settings where diagnostic resources are not available as it is cheap, easy to implement and offers same-day treatment [1]. However, syndromic management for vaginal discharge is associated with unnecessary use of antibiotics which fuel the development of antimicrobial resistance (AMR), and importantly it leaves asymptomatic infections untreated [2].

Although WHO guidelines now include provision for aetiological testing, there are barriers to operationalising this in resource-limited settings [3]. Ideally, diagnostics should meet the ASSURED criteria (affordable, sensitive, specific, user-friendly, rapid, robust, easy-to-use, deliverable) [4]. Previously developed non-molecular tests for *Neisseria gonorrhoeae* have not met the minimum requirements for implementation [5–7] Nucleic acid amplifications tests (NAATs) are considered the current gold standard assay for detection of NG with several commercial assays available. However, cost and time-to-result limit their use in resource-limited settings.

*Neisseria gonorrhoeae* (NG) is associated with pelvic inflammatory disease, infertility, adverse birth outcomes, and increased risk of HIV transmission [8], and has been identified as a WHO priority pathogen for AMR research and development initiatives [9]. Improving antimicrobial stewardship for NG, by reducing unnecessary antibiotic use, requires diagnostic tests that can differentiate between individuals with and without infection.

A novel lateral flow assay for *N. gonorrhoeae* (NG-LFA) with an analytic time of 20 minutes, was reported to have sensitivities of 96% and 92%, and specificities of 97% and 96% among symptomatic men and women, respectively, in South Africa [10] and was found to be acceptable and usable for clients and health workers [11]. However, performance in the context of aetiological management cannot be extrapolated to screening amongst asymptomatic individuals. Clinical test performance may differ if bacterial loads are lower among asymptomatic than symptomatic individuals whilst the predictive values for both negative and positive results will be affected by the prevalence among different populations. Finally, perceptions of the utility of such a diagnostic as a screening tool rather than an aetiological test may differ, with implications for deployment within health services.

STI screening, particularly for NG, could be particularly effective among pregnant women. Both observational and interventional studies have demonstrated an association between NG in pregnancy and adverse birth outcomes such as prematurity and low birth weight, the incidence of which could be reduced by NG screening and treatment [12–14] An NG-LFA with high specificity and good sensitivity could potentially facilitate adoption of NG screening in pregnant women in resource constrained settings.

The aim of this study was to evaluate the performance and acceptability of the NG-LFA as a screening test in pregnant women, against molecular testing using the GeneXpert platform.

## Methods

### Study design and setting

This validation study was embedded within a prospective interventional study evaluating the implementation of STI screening for pregnant women in Harare, Zimbabwe. The parent study protocol has been published [15]. The study was conducted in two high volume antenatal

care (ANC) clinics in urban, high-density settings in southwest Harare. In this setting routine ANC is embedded within primary healthcare clinics and includes midwife-led deliveries, with referral to hospital for complicated deliveries.

## Study population

The study population consisted of pregnant women attending routine ANC. Exclusion criteria were prior enrolment into the study or being unable to provide informed consent. Participants were not excluded based on ANC visit number, gestational age, or symptom status. Participants were enrolled sequentially, as described in the parent study protocol [15].

## Study procedures

Participants were enrolled between 9th February 2023 and 23rd October 2023. Following written informed consent, three nurse-collected vaginal swabs were collected in the following order for testing for: 1) *Trichomonas vaginalis* (TV) using the OSOM Trichomonas Rapid Test (Sekisui); 2) *Chlamydia trachomatis* (CT) and NG using the Xpert CT/NG assay (Xpert) on the GeneXpert platform (Cepheid); and 3) NG testing using the NG-LFA. Swabs were collected with participants in the lithotomy position (without stirrups), with swabs inserted 5 cm inside the opening of the vagina and gently rotated for 10 seconds.

The NG-LFA has been described previously [10,11,16]. It is a single-use disposable lateral flow assay that uses a fluorescent europium reporter for the detection of NG during a patient visit. Fluorescence is detected by a small portable point-of-care reader that classifies results into positive, negative, or invalid, within 20 minutes using simple LED light indicators. The fluorescence threshold for a positive reading was ≥0.063 [10]. For offline analysis, fluorescence intensity data for each test can be downloaded from the reader. Positive and negative quality control tests were run on readers daily before testing.

For invalid NG-LFA results, a new vaginal swab was immediately collected for repeat testing with a new test. For invalid Xpert results, repeat testing was performed using excess transport reagent from the tube containing the swab. If insufficient reagent remained, a new vaginal swab was collected.

Management for NG infection was only provided for positive results from the reference Xpert test, and included antibiotics, risk reduction counselling, and partner notification and treatment in line with national guidelines [17]. For participants positive for NG on either Xpert or the NG-LFA, an additional cervical swab was collected for gonococcal culture. Due to the differing analytic times, Xpert results were available after the NG-LFA results, ensuring "blinding" of the study team to NG-LFA interpretation.

Sociodemographic and clinical data was collected from participants using an interviewer-administered questionnaire using Open Data Kit on tablets.

Cycle threshold values for Xpert-positive results were extracted from the GeneXpert output for both NG2 and NG4 targets, which are two separate chromosomal targets for NG, that are amplified as part of results processing [18]. Both NG2 and NG4 targets must be detected for the Xpert result to be reported as positive.

## Laboratory procedures

Cervical samples were collected using the eSwab collection and transport system (Copan, Italy). These were transported to the Biomedical Research and Training Institute laboratory, where they were plated on InTray GC (Biomed Diagnostics) in vitro devices. A sterile loop was used to streak observed colonies on chocolate agar culture media, which was incubated at 36 ± 1°C in an atmosphere supplemented with 5% $CO_2$. Plates were examined 24-48 hours after inoculation.

NG was presumptively identified by the presence of all of; 1) morphologically typical colonies; 2) typical gram-negative diplococci on gram staining; 3) positive oxidase testing; and 4) positive superoxol testing, in sub-cultured isolates.

## Acceptability and usability

The four members of the implementation team (two nurses and two research assistants) completed the system usability scale (SUS) at the end of the intervention period [19]. The SUS is a validated, 10-item questionnaire with a 5-point (strongly agree to strongly disagree) Likert scale to measure perceived usability, with questions pertaining to confidence in use, ease of learning, and system complexity [19]. The final score is out of 100. Scores above 68 and above 80.3 were considered acceptable and excellent, respectively [11].

In-depth interviews were also conducted with these four members as part of the broader intervention process evaluation. Interviews were conducted by an interviewer trained in qualitative interviewing not involved in service delivery, using a topic guide. Interviews were conducted at the study sites, one week after the end of participant recruitment, and lasted between 27-65 minutes.

The SUS was performed just prior to the interviews to stimulate discussion. Additionally, focussed topic guide questions covered acceptability and usability of the NG-LFA, including results interpretation, reader design, durability, reliability, and technical issues.

## Analysis

Data was analysed using STATA version 18.0 (StataCorp, Texas, USA). Sample size was based on the parent study [15].

We calculated the sensitivity, specificity, positive predictive value (PPV), negative value (NPV), and accuracy of the NG-LFA assay taking the Xpert as the reference diagnostic. Exploratory descriptive analyses were conducted to assess these measures according to symptom status, polymerase chain reaction (PCR) cycle threshold (Ct) value, antibiotic usage, HIV status, age, trimester, clinic site, and NG-LFA reader. The Mann–Whitney U test was used to compare PCR Ct values of Xpert positive samples detected and not detected by the NG-LFA, and between individuals with and without symptoms.

A receiver operating characteristic (ROC) curve, and the area under the curve (AUC), was generated from the NG-LFA fluorescence values, with Xpert as the reference standard. The Youden index was also calculated, using the "cutpt" command in Stata, which is the threshold value representing the optimal compromise between sensitivity and specificity, giving equal weight to each [20].

Interview data was analysed using thematic analysis. Open coding was performed by KM in Microsoft Word. Codes were grouped together inductively to develop themes, which were then reviewed, named, and defined. Four main themes were developed in the interviews; 1) Ease of use and interpretation; 2) Positive device characteristics; 3) Discrepancies with a seemingly infallible gold standard, leading to mistrust of results; and 4) Importance of a back-up test during future use.

## Ethics

Ethical approval for the study was obtained from the Medical Research Council of Zimbabwe (MRCZ/A/2899) and the London School of Hygiene & Tropical Medicine Interventions Ethics Committee (26787). All participants provided written informed consent in Shona or English. In Zimbabwe, individuals who are under 18 years of age and pregnant are considered emancipated minors. Therefore, independent informed consent was obtained in the same

manner from pregnant minors as for adults. Results have been reported according to Studies of Diagnostic Accuracy (STARD) guidelines [21].

### Role of the funding source

The funders of the study had no role in study design, data collection, data analysis, data interpretation, or writing of the report.

## Results

### Participant characteristics

Study recruitment is detailed in the flow diagram in Fig 1, with 912 individuals included in the final analyses. The median participant age was 25 (IQR 22 – 30) years. Most participants were attending ANC for a booking visit (71.5%; 652/912) and 27.9% (254/912) were attending for their first pregnancy. Of 815 participants with an estimated gestational age, 18 (2.2%), 350 (42.9%), and 447 (54.9%) were in the first, second, and third trimesters, respectively.

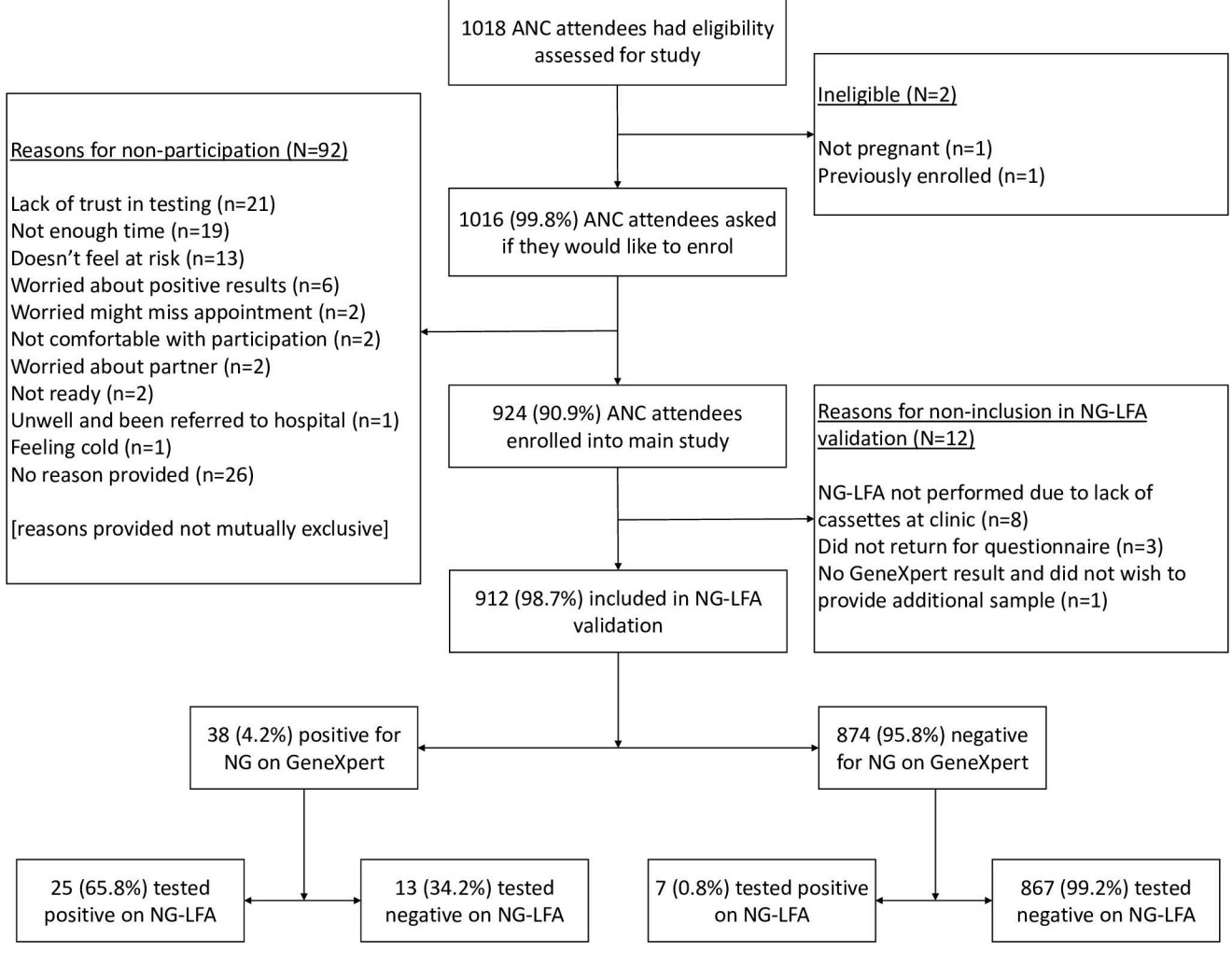

**Fig 1. Flow diagram.**

HIV prevalence was 9.9% (90/912), of whom 17 (18.9%) individuals were newly diagnosed. All 73 who reported HIV positive status prior to testing also reported antiretroviral therapy use. CT and TV prevalence was 18.8% (171/912) and 11.7% (107/912), respectively.

Most individuals (91.7%) reported no genital symptoms. The most commonly reported symptoms were abnormal vaginal discharge (4.8%; 44/912), vulval irritation or itching (3.3%; 30/912), and lower abdominal or pelvic pain (1.2%; 11/912). Only 2.0% (18/912) reported antibiotic use within the preceding two weeks, of whom four individuals reported receiving ceftriaxone (Table 1).

## Performance of the NG-LFA

The prevalence of NG by Xpert was 4.2% (38/912). The number of true positives, true negatives, false positives, and false negatives using the NG-LFA compared to the reference Xpert assay was 25 (2.7%), 867 (95.1%), 7 (0.8%), and 13 (1.4%), respectively. The sensitivity, specificity, PPV, and NPV of the NG-LFA were 65.8% (25/38; 95% CI 48.6%–80.4%), 99.2% (867/874; 95% CI 98.4–99.7%), 78.1% (25/32; 95% CI 60.0-90.7%), and 98.5% (867/880; 95% CI 97.5-99.2%). Test accuracy was 97.8% (892/912; 95% CI 96.6-98.7%).

Sensitivity and specificity, disaggregated by symptom status, Ct value, antibiotic usage, HIV status, age, gestational age, clinic site, and reader, is shown in Table 2. For all eighteen individuals who reported antibiotic use within the previous two weeks, both Xpert and NG-LFA were negative.

There were six "invalid" NG-LFA results on initial testing, all of which on repeat testing were negative. All six were also negative on Xpert. Of the 912 enrolled participants, 17 had error codes reported on Xpert. Repeat testing returned negative results for all participants bar one who was positive. One further participant was excluded from analyses following an error code, as they did not wish to provide another sample for re-testing.

No adverse events were reported from performing either the NG-LFA or Xpert.

## NG-LFA performance across cycle threshold values and within sub-groups

Figs 2 and 3 show the distribution of Xpert NG2 and NG4 target Ct values in Xpert positive samples, disaggregated by NG-LFA positivity. S1 Fig shows a scatter plot of NG2 and NG4 Ct values, disaggregated by NG-LFA positivity.

The Ct values for both NG2 (median 29.5 vs 25.5; p = 0.0001) and NG4 (median 30.2 vs 25.6; p=0.0002) targets were higher in participants with false negative NG-LFA results, compared to true positive. Of the 13 discordant NG-LFA negative results, 8 had an NG2 and/or NG4 PCR Ct value of above 30; the remaining five had NG2 and NG4 Ct values between 25 and 30.

Among participants with positive Xpert results, 18.4% (7/38) self-reported presence of vaginal discharge. There was no difference in Ct values between those with and without symptoms, for both NG2 (median 26.3 vs 26.2; p = 0.90) and NG4 (median 26.3 vs 26.3; p=0.79) targets. S1 Table shows the characteristics of participants positive for NG on Xpert, disaggregated by NG-LFA positivity.

## ROC curve

The ROC curve is shown in Fig 4. The AUC was 0.94 (95% CI 0.89 – 0.98). The optimal cut point according to the Youden Index was ≥0.0278785, which provided a sensitivity, specificity, PPV, and NPV, at this cut point of 89.5% (95% CI 75.2 – 97.1%), 90.4% (95% CI 88.2 – 92.2%), 28.8% (95% CI 20.8 – 37.9%), and 99.5% (95% ci 98.7 – 99.9%), respectively.

Table 1. Characteristics of participants recruited into the study (N = 912 unless otherwise stated).

| Variable | N (%) |
|---|---|
| **Sociodemographic** | |
| **Age (years)** | |
| 15 – 19 | 121 (13.3%) |
| 20 – 24 | 309 (33.9%) |
| 25 – 29 | 231 (25.3%) |
| ≥30 | 251 (27.5%) |
| **Clinic** | |
| Site A | 613 (67.2%) |
| Site B | 299 (32.8%) |
| **Education level** | |
| Primary or below | 78 (8.6%) |
| Secondary | 788 (86.4%) |
| Vocational | 11 (1.2%) |
| Higher/University | 35 (3.8%) |
| **Current employment status** | |
| Unemployed | 474 (52.0%) |
| Student | 14 (1.5%) |
| Self-employed/ business owner | 114 (12.5%) |
| Salaried employment | 69 (7.6%) |
| Informal work | 241 (26.4%) |
| **Relationship with father of child** | |
| Married | 502 (55.0%) |
| Not married, but living together | 335 (36.7%) |
| Not married or living together | 45 (4.9%) |
| No relationship | 30 (3.3%) |
| **Pregnancy history** | |
| **Antenatal care visit type** | |
| Booking | 652 (71.5%) |
| Follow-up | 260 (28.5%) |
| **Trimester (N = 815)** | |
| First | 18 (2.2%) |
| Second | 350 (42.9%) |
| Third | 447 (54.9%) |
| **Number of previous pregnancies** | |
| 0 | 254 (27.9%) |
| 1 | 246 (27.0%) |
| 2 | 204 (22.3%) |
| ≥3 | 208 (22.8%) |
| **Clinical status** | |
| **Antibiotic use in previous two weeks** | |
| Yes | 18 (2.0%) |
| No | 892 (97.8%) |
| Unknown | 2 (0.2%) |
| **Antibiotics used in previous two weeks (N = 18)** | |
| Amoxicillin only | 4 (22.2%) |
| Metronidazole only | 2 (11.1%) |
| Ceftriaxone/ Erythromycin | 2 (11.1%) |

*(Continued)*

**Table 1.** (Continued)

| Variable | N (%) |
|---|---|
| Azithromycin only | 1 (5.6%) |
| Erythromycin only | 1 (5.6%) |
| Ceftriaxone/ Azithromycin | 1 (5.6%) |
| Ceftriaxone/ Erythromycin/ Metronidazole | 1 (5.6%) |
| Amoxicillin/ Erythromycin | 1 (5.6%) |
| Amoxicillin/ Erythromycin/ Benzathine penicillin | 1 (5.6%) |
| Amoxicillin/ Metronidazole | 1 (5.6%) |
| Erythromycin/ Metronidazole | 1 (5.6%) |
| Cannot recall | 2 (11.1%) |
| **Current symptoms**\* | |
| Abnormal vaginal discharge | 44 (4.8%) |
| Vulval irritation or itching | 30 (3.3%) |
| Lower abdominal or pelvic pain | 11 (1.2%) |
| Genital ulcer, sore, or swelling | 8 (0.9%) |
| Dyspareunia | 4 (0.4%) |
| Other# | 13 (1.4%) |
| No symptoms | 836 (91.7%) |

\*Symptoms not mutually exclusive.

#Other symptoms: warts (n=6), rash (n=4), rectal pain (n=2), foul smell (n=1)

## Gonococcal culture

Of the 38 Xpert results positive for NG, 14 (36.8%) were cultured and identified as NG by the criteria detailed in the methods. Of the 13 discordant false negative results by NG-LFA, 3 (23.1%) were culture positive. Of the 7 'false positive' NG-LFA results, 6 participants provided a cervical swab, of which one (16.7%) was cultured and identified as NG.

## Acceptability and usability

All staff reported the NG-LFA was usable with SUS scores of 70 and 85 for the two research assistants, and 85 and 90 for the two nurses.

The NG-LFA was considered "*very user-friendly… simple to follow*" (research assistant), "*easy to use*" (nurse), and the training provided was considered adequate. The reader was reported to be "*portable and very easy to carry*" (nurse), with a good battery life, and having a short time to results: "*30 minutes to have results, haa that's wonderful*" (research assistant). Overall, the reader was not considered a barrier to use of the NG-LFA.

Despite this, a recurrent theme was a mistrust of the NG-LFA because of "*discrepancies with its results*" (research assistant) with Xpert, which was generally treated as infallible. Any discrepancies were considered to be due to a failure of the NG-LFA: "*the results on the flow test were not reliable*" (research assistant). Importantly, the team were aware of both sets of results which allowed for this mistrust to develop. Supporting quotes are shown in S2 Table.

## Discussion

In this largely asymptomatic pregnant population, we demonstrated a very high specificity and NPV of the NG-LFA. However, the sensitivity was lower than reported in a symptomatic general population [10]. One in three cases of NG were not detected by the NG-LFA among pregnant women. Based on the findings of this study the NG-LFA potentially provides value

**Table 2.  Sensitivity and specificity of novel lateral flow assay for *Neisseria gonorrhoeae* infection, in pregnant women (N=912 unless otherwise stated).**

| Variable | Sensitivity % (n/N) | 95% CI | Specificity % (n/N) | 95% CI |
|---|---|---|---|---|
| Overall | 65.8% (25/38) | 48.6 - 80.4% | 99.2% (867/874) | 98.4 – 99.7% |
| Vaginal discharge | | | | |
| Present | 57.1% (4/7) | 18.4 – 90.1% | 91.9% (34/37) | 78.1 – 98.3% |
| Absent | 67.7% (21/31) | 48.6 – 83.3% | 99.5% (833/837) | 98.8 – 99.9% |
| Cycle threshold (NG2) | | | | |
| <30 | 75.8% (25/33) | 57.7 – 88.9% | – | – |
| >30 | 0.0% (0/5) | 0 – 52.2%* | – | – |
| Cycle threshold (NG4) | | | | |
| <30 | 82.8% (24/29) | 64.2 – 94.2% | – | – |
| >30 | 11.1% (1/9) | 0.3 – 48.2% | – | – |
| Antibiotic usage in previous 2 weeks (N = 910) | | | | |
| Yes | – | – | 100.0% (18/18) | 81.5 – 100.0%* |
| No | – | – | 99.2% (847/854) | 98.3 – 99.7% |
| HIV status | | | | |
| Positive | 83.3% (5/6) | 35.9 – 99.6% | 100.0% (84/84) | 95.7 – 100.0%* |
| Negative | 62.5% (20/32) | 43.7 – 78.9% | 99.1% (783/790) | 98.2 – 99.6% |
| Age | | | | |
| 15 – 19 | 64.3% (9/14) | 35.1 – 87.2% | 100.0% (107/107) | 96.6 – 100.0%* |
| 20 – 24 | 71.4% (5/7) | 29.0 – 96.3% | 99.0% (299/302) | 97.1 – 99.8% |
| 25 – 29 | 66.7% (4/6) | 22.2 – 95.7% | 99.6% (224/225) | 97.5 – 100.0% |
| 30 + | 63.6% (7/11) | 30.8 – 89.1% | 98.8% (237/240) | 96.4 – 99.7% |
| Trimester (N=815) | | | | |
| First | 50.0% (1/2) | 1.3 – 98.7% | 100.0% (16/16) | 79.4 – 100.0%* |
| Second | 63.6% (7/11) | 30.8 – 89.1% | 99.4% (337/339) | 97.9 – 100.0% |
| Third | 65.2% (15/23) | 42.7 – 83.6% | 99.3% (421/424) | 97.9 – 99.9% |
| Site | | | | |
| Site A | 60.7% (17/28) | 40.6 – 78.5% | 99.3% (581/585) | 98.3 – 99.8% |
| Site B | 80.0% (8/10) | 44.4 – 97.5% | 99.0% (286/299) | 92.7 – 97.7% |
| Reader (N=910) | | | | |
| Reader 1 (Site B) | 87.5% (7/8) | 47.3 – 99.7% | 100.0% (140/140) | 97.4 – 100.0%* |
| Reader 2 (Site A) | 53.9% (7/13) | 25.1 – 80.8% | 99.1% (215/217) | 96.7 – 99.9% |
| Reader 3 (Site A) | 66.7% (10/15) | 38.3 – 88.2% | 99.5% (365/367) | 98.0 – 99.9% |
| Reader 4 (Site B) | 50.0% (1/2) | 12.6 – 98.7% | 98.0% (145/148) | 94.2 – 99.6% |

*one sided, 97.5% confidence interval.

in this context by allowing identification of asymptomatic women with NG who would otherwise be untreated, and may serve as a useful screening tool to rule in and treat NG infection in pregnant populations. However, more consideration is required when using the NG-LFA to rule out NG infection in ANC clinics, which should be supported by further demonstration studies. Modelling is also required to determine whether this level of clinical performance is still cost-effective compared to current standard of care, which is no screening.

Among Xpert-detected NG cases negative by NG-LFA, higher Ct values suggest that lower bacterial load in these samples is the likely reason for non-detection [22]. Other possibilities for false negative results include human error or issues with the readers. The four readers used in the study were prototype models and could have had technical issues. There were four false

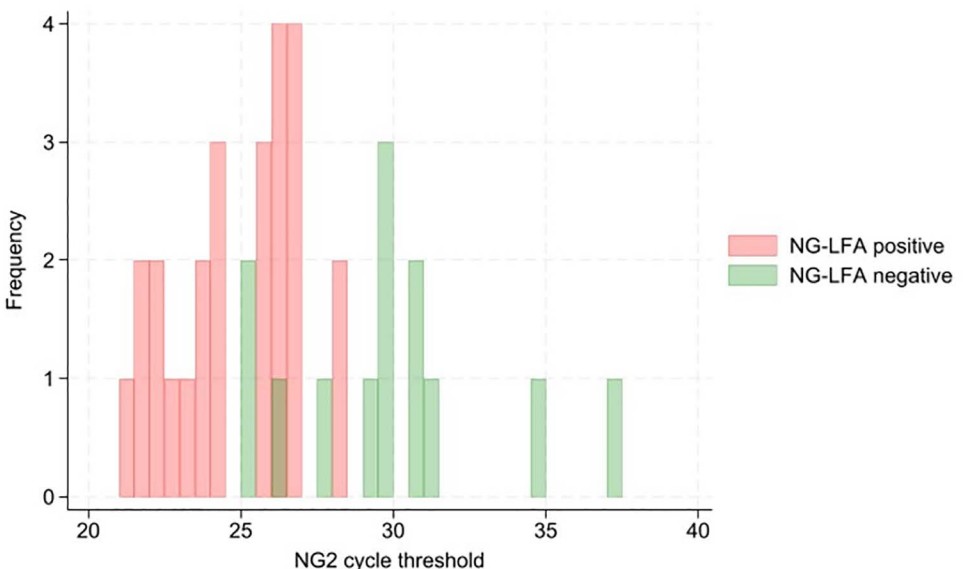

**Fig 2. Histogram demonstrating distribution of Xpert NG2 target cycle threshold values in Xpert positive samples (N = 38).**

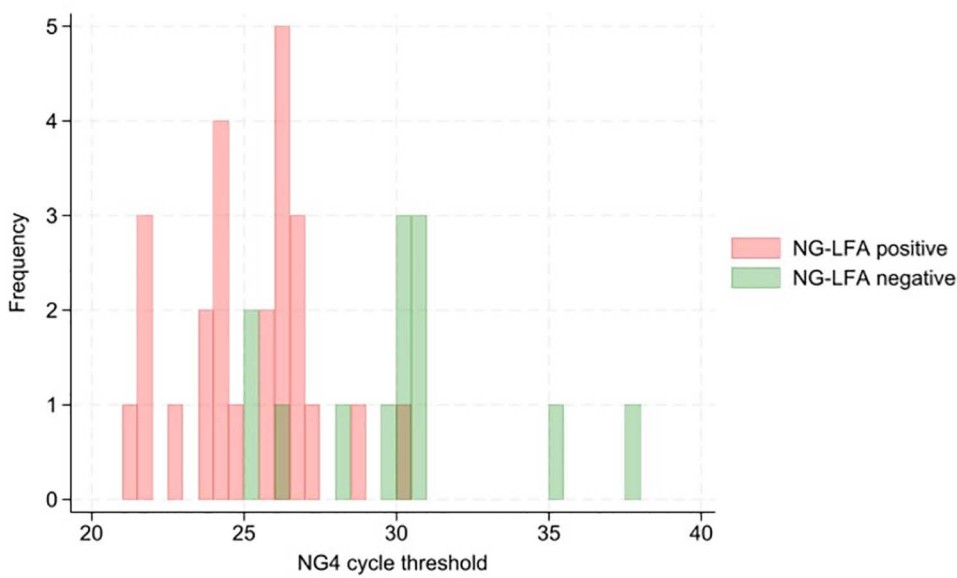

**Fig 3. Histogram demonstrating distribution of Xpert NG4 target cycle threshold values in Xpert positive samples (N = 38).**

negative results within eight days at one site in August 2023. However, prior to re-training, practice tests using positive and negative control samples were correctly identified by both readers at the site. As no obvious errors could be identified retrospectively, and the relatively high Ct values among the false negative cases in this period (NG4 Ct > 30 for three out of four cases), this may have been a random increase in the frequency of false negative cases. There was also no difference in sensitivity between sites and readers, but confidence intervals were

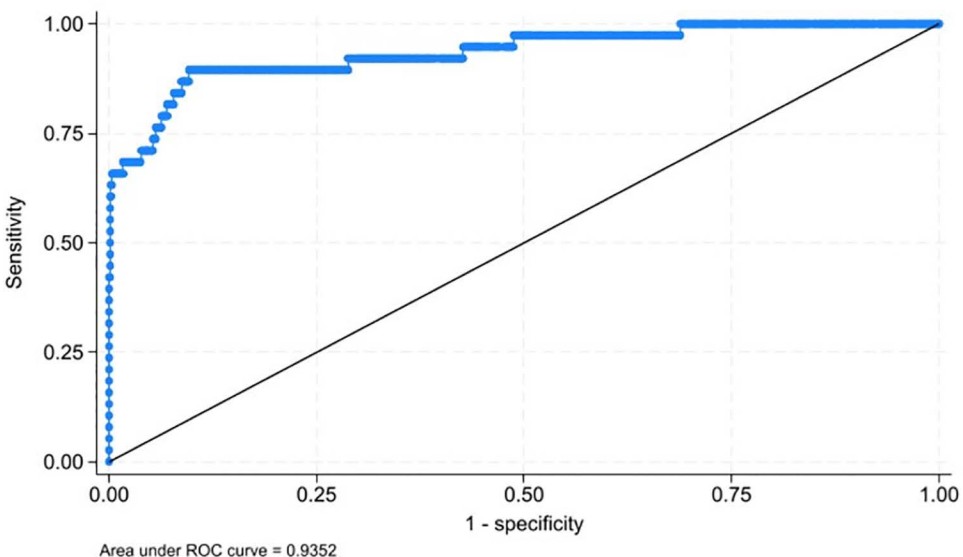

**Fig 4.  Receiver Operating Characteristic (ROC) curve of Neisseria gonorrhoeae lateral flow assay (NG-LFA) at different fluorescence thresholds.**

wide. Strengthening and ensuring quality assurance of the reader will be particularly important prior to commercialisation of the NG-LFA. Another consideration that may have contributed to lower bacterial loads was that the NG-LFA specimen was the third swab collected. In contrast, in the previous evaluation by Peters et al. the first swab collected was tested using the NG-LFA [10].

The previous evaluation of the NG-LFA focused on improving aetiological diagnosis and management [10]; in this study we evaluated its use for screening. The goals of testing vary markedly across these scenarios. In symptomatic patients, NG testing aims to prevent inappropriate treatment for NG in individuals who would otherwise receive treatment as part of syndromic management. In contrast, the goal of screening for NG in ANC would be to detect and treat NG cases that would otherwise not be treated, to avoid downstream sequelae, whilst also not providing inappropriate treatment to individuals without infection. The individual level benefits conferred by a test with a sensitivity of 65% and very high specificity may therefore be acceptable when compared to the alternative of no testing, until low-cost point-of-care tests with higher sensitivity are developed.

The evidence for screening for asymptomatic NG infection in pregnancy is not conclusive. The WANTAIM trial found that among pregnant women with NG at enrolment, screening and treatment led to a 53% improvement in prevalence of preterm birth and/or low birth weight, although this was in a small subset of trial participants [14]. Additionally, given that ophthalmia neonatorum is mainly contracted during delivery due to presence of NG in the birth canal, antenatal NG treatment should reduce this risk [23]. However, the clinical importance of the NG cases detected by Xpert but missed by the NG-LFA in this study is unclear. Lateral flow antigen tests have a lower analytical sensitivity than NAATs, which can detect prior or resolving infections, typically with high Ct, where DNA may be detectable but not antigen [24–26] In some cases, high Ct may also represent results that are falsely positive on Xpert, which is not a perfect diagnostic [27]. Previous studies have reported spontaneous clearance rates of up to 33% for NG in non-pregnant populations [28,29], Hypothetically, some low bacterial load infections might self-clear and the risks of treatment may outweigh

the benefits. Understanding the downstream consequences of treating or not treating such infections is essential to determine the clinical and cost-effectiveness of screening. Of note, there were three false negative results that were culture positive, indicative of viability. This discrepancy may due to the different sampling sites for the NG-LFA and for culture. Whereas swabs for NG-LFA were vaginal, swabs for culture were taken from the cervix, where both bacterial load and organism viability is likely to be higher.

We compared sensitivity and specificity across a range of participant characteristics. However, due to the small sample sizes for sensitivity calculations, it is difficult to draw meaningful conclusions from these observations. Caution must also be exercised when comparing these findings to the study among symptomatic individuals in South Africa [10]. Individuals attending clinic because they have symptoms are likely very different to those attending for routine ANC and reporting symptoms when specifically asked. Firstly, physiological changes to vaginal discharge are common in pregnancy. Furthermore, the former's symptoms are more likely to be of a higher severity or duration such that the individual has sought care. For these reasons we are also wary of interpreting the lack of difference in Ct values between participants with and without symptoms.

Such differences between populations are also important when considering the fluorescence threshold at which a sample is considered NG positive, and how optimal thresholds will vary between different populations. In both this study and the evaluation among symptomatic patients in South Africa, the threshold for a positive reading was ≥0.063. However, in our study a reduced threshold of ≥0.0278785 would have led to a sensitivity and specificity of 89.5% and 90.4%, respectively. By sacrificing a degree of specificity, only 4 NG infections out of 38 would have been missed. However, one in ten of those without NG would have been inappropriately treated. This demonstrates the importance of the aim of screening or testing in determining the most appropriate balance between sensitivity and specificity. For example, if the primary goal of ANC NG screening is to avert cases of ophthalmia neonatorum, then a relatively high rate of overtreatment may be tolerated. Finally we calculated the Youden index, which gives equal weight to sensitivity and specificity [20]. However, a different weighting may be used to find the most appropriate threshold. Importantly, as the NG-LFA has not yet gone through regulatory approval and is not commercialised, the threshold in both this study and the evaluation among symptomatic individuals was based on limited data. The threshold for a commercialised assay would be based on larger datasets covering different populations.

The NG-LFA was reported to be acceptable, easy-to-use, easy-to-interpret, and portable, concurrent with the SUS scores. These characteristics are particularly important when considering implementation in settings with limited infrastructure and where retention of a skilled workforce is challenging. However, interviewees reported issues with trusting the device following their experience of discordant results. Suggestions from team members of potential uses for the NG-LFA tended to include the need for a back-up, such as confirmatory testing, or the provision of presumptive treatment in individuals with persistent symptoms but a negative NG-LFA result. In contrast, in South Africa the NG-LFA was reported to engender trust among both healthcare workers and clients, as they could see the test being performed in front of them, in addition to providing increased confidence in clinical decision making [30]. However, there are some key differences between the studies in this regard. In South Africa, the aim was to rule out NG infection and reduce antimicrobial usage in symptomatic individuals who would have otherwise received treatment, and so having results to inform this management is positive. Furthermore, because of the higher sensitivity and higher NG prevalence, the impact of false positive and false negative results is reduced. In our study, the NG-LFA correctly reported 97.8% of results, but the small number of incorrect results had a notable effect on its perceived reliability. Another consideration is how the NG-LFA was

communicated across both environments. In our study, Xpert was already integrated into ANC when the NG-LFA was introduced, and was used to guide treatment. The Xpert results were therefore considered the "gold standard". In South Africa, testing procedures included additional tie breaker tests where discordant results were recorded, suggesting to staff that Xpert may not always be correct [10].

The strengths of this study include its pragmatic design, with participants attending for routine ANC within public sector primary healthcare clinics in Harare, reflecting real-world use of the NG-LFA. There were also important limitations. As the number of women with positive Xpert results was relatively small, the confidence intervals for the sensitivity estimates are wide, limiting comparisons of test performance across different participant characteristics. Importantly, the overall study sample size was not powered for the present evaluation, and was likely underpowered. Additionally, we used Xpert as the reference standard. Although a molecular test with high sensitivity and specificity [31], in the previous evaluation, laboratory-based confirmatory PCR was performed on discordant cases and identified that Xpert had three false negative and two false positive results [10]. Given the lower pre-test probability for screening in our study, false positive results on the Xpert are even more likely [25,32]. Furthermore, one of the "false positive" NG-LFA infections in this study was cultured and identified phenotypically as NG, indicating a false negative Xpert result. Unfortunately, overall yield from culture of NG isolates was relatively poor, with just over a third of individuals with Xpert-positive NG having a positive culture result. Although NG culture was never intended to be the gold standard, it was still less helpful than had been envisioned, particularly in terms of confirming false positive or false negative results. The lack of an on-site incubator to allow for immediate plating, and also variable times between sample collection and transport to the laboratory may have been contributing factors. Finally, due to the limited number of interviewees and limited time discussing the NG-LFA within broader interviews, our assessment of acceptability and usability is based on only four viewpoints. Furthermore, although the intervention team were blinded to Xpert results at the time of reading the NG-LFA, they did subsequently find out the Xpert results, due to the interventional nature of the study and having to deliver treatment to participants. This undoubtedly influenced their perceptions of the reliability of the NG-LFA. Of note, if such a test were implemented at scale, concerns regarding accuracy or trust are unlikely to be realised at a local level, if such confirmatory tests are not performed.

In conclusion, although the NG-LFA is promising for augmenting syndromic management in order to prevent inappropriate antimicrobial usage, as demonstrated among symptomatic individuals in South Africa [10], the potential usability of the NG-LFA as a screening tool in this context is less evident. NPV was high, and so the NG-LFA would correctly rule out infection in the majority of individuals. However, one-third of pregnant women with Xpert-detected NG were missed, which may have important implications for clinical and cost-effectiveness, and performance may not be considered acceptable for widespread implementation. However, this must be weighed against the current standard of care relying on syndromic management alone. Furthermore, most discordant NG-LFA negative cases had high Ct values suggestive of low bacterial load. More studies are needed to assess the NG-LFA in other settings and populations, across different sample types and sample collection methods, to further explore the relationship between NG-LFA positivity, bacterial load and infection viability, and to understand the clinical implications of these low bacterial load infections. Current NAATs are too expensive and/or logistically unfeasible for routine care, and further research is warranted to develop highly sensitive screening tests for asymptomatic pregnant women.

## Supplementary information

**S1 Fig. Scatter plot of NG2 and NG4 cycle threshold values, disaggregated by NG-LFA positivity.**
(TIF)

**S1 Table. Characteristics of participants positive for *Neisseria gonorrhoeae* on Xpert, comparing those with NG-LFA positive and negative results (N=38 unless otherwise stated).**
(DOCX)

**S2 Table. Themes and supporting quotes from interviews.**
(DOCX)

**S1 Checklist. STARD checklist.**
(DOCX)

## Author contributions

**Conceptualization:** Kevin Martin, Victoria Simms, Laura T. Mazzola, Benjamin Blumel, Birgitta Gleeson, Michael Marks, Katharina Kranzer, Rashida Ferrand, Cecilia Ferreyra.

**Data curation:** Kevin Martin, Tsitsi Bandason.

**Formal analysis:** Kevin Martin, Constance R. S. Mackworth-Young, Victoria Simms.

**Funding acquisition:** Kevin Martin.

**Investigation:** Kevin Martin, Beauty Makamure, Janice Martin, Forget Makoga.

**Methodology:** Kevin Martin, Chido Dziva Chikwari, Anna Machiha, Remco P. H. Peters.

**Project administration:** Kevin Martin, Ethel Dauya, Laura T. Mazzola, Benjamin Blumel, Birgitta Gleeson, Cecilia Ferreyra.

**Resources:** Beauty Makamure.

**Supervision:** Kevin Martin, Michael Marks, Katharina Kranzer, Rashida Ferrand.

**Visualization:** Kevin Martin.

**Writing – original draft:** Kevin Martin.

**Writing – review & editing:** Kevin Martin, Ethel Dauya, Chido Dziva Chikwari, Constance R. S. Mackworth-Young, Victoria Simms, Tsitsi Bandason, Beauty Makamure, Janice Martin, Forget Makoga, Anna Machiha, Remco P. H. Peters, Laura T. Mazzola, Benjamin Blumel, Birgitta Gleeson, Michael Marks, Katharina Kranzer, Rashida Ferrand, Cecilia Ferreyra.

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
