## [Decision Letter · Decision Letter 0]

17 Sep 2024

PGPH-D-24-01660

Evaluation of a novel point-of-care lateral flow assay screening for Neisseria gonorrhoeae infection among pregnant women in Zimbabwe

Dear Dr. Martin,

Thank you for submitting your manuscript to PLOS Global Public Health. After careful consideration, we feel that it has merit but does not fully meet PLOS Global Public Health’s publication criteria as it currently stands. Therefore, we invite you to submit a revised version of the manuscript that addresses the points raised during the review process.

We look forward to receiving your revised manuscript.

Kind regards,

Gagandeep Singh, M.D.

Academic Editor

Journal Requirements:

1. We do not publish any copyright or trademark symbols that usually accompany proprietary names, eg (R), (C), or TM  (e.g. next to drug or reagent names). Please remove all instances of trademark/copyright symbols throughout the text, including ® on pages 5 and 17.

The article may be revised as per the suggestions of the reviewers and resubmitted.

Reviewers' comments:

Reviewer's Responses to Questions

**Comments to the Author**

1. Does this manuscript meet PLOS Global Public Health’s publication criteria ? Is the manuscript technically sound, and do the data support the conclusions? The manuscript must describe methodologically and ethically rigorous research with conclusions that are appropriately drawn based on the data presented.

Reviewer #1: Yes

Reviewer #2: Yes

2. Has the statistical analysis been performed appropriately and rigorously?

Reviewer #1: No

Reviewer #2: Yes

3. Have the authors made all data underlying the findings in their manuscript fully available (please refer to the Data Availability Statement at the start of the manuscript PDF file)?

Reviewer #1: Yes

Reviewer #2: Yes

4. Is the manuscript presented in an intelligible fashion and written in standard English?

Reviewer #1: Yes

Reviewer #2: Yes

5. Review Comments to the Author

Reviewer #1: There is an urgent need for POCTs for STIs. This article on the use of NG-LFA as a screening tool in pregnant women in Zimbabwe NG highlights important data and recent developments in the field. The paper is well written. The authors have captured both, the advantages as well as limitations of the point-of care test for screening for NG in setting which follows the syndromic management of STIs. This assay has already demonstrated a high sensitivity and specificity (>90%) in symptomatic individuals. It was only logical to next evaluate its performance & acceptability as a screening tool.

I have minor comments for the authors :

1. In the Laboratory procedures, the authors have not carried out confirmatory testing for NG. The tests mentioned only provide presumptive identification of NG.

2. The false-negative results by NG-LFA were seen basically in samples that had a low bacterial load. We do expect a low load in asymptomatic patients. Subsequently, it is mentioned that there was no difference in Ct values between those with and without symptoms, for both NG2 (median 26.3 vs 26.2; p = 0.90) and NG4 (median 26.3 vs 26.3; p=0.79) targets. Please explain.

3. Gonococcal culture: Of the 13 discordant results by NG-LFA, 2 (15.4%) were culture positive. This is a matter of serious concern. What is the limit of detection of NG-LFA ?

4. Some issues can be explained by the fact that the third swab was used for NG-LFA.

5. One third of pregnant women were missed by NG-LFA. This has important implications particularly, involving the neonate. No doubt, we need screening tools with better sensitivity and more research is needed in this direction.

Reviewer #2: 1. The study is part of larger approved size for which the sample size calculation was done. Hence, the sample size seems likely underpowered for the present study. Eventhough drawing meaningful interpretations from current sample size is difficult; the results available for the current sample size are well presented and interpreted highlighting the existing limitations.

2. On its own the study is more suitable to be presented as a pilot work, based upon whose results a comprehensive study to evaluate the LFA may be planned including adequate sample size for evaluating both the performance characteristics and operational characteristics of the LFA considering:

a. Both symptomatic/ asymptomatic participants

b. High risk population/ general population

c. Types of samples including (vaginal swab/ urine/ rectal swab/oral swab/ cervical swabs etc.)

d. self collected/ clinician collected samples

This will present a correct picture of the utility of the LFA especially in resource limited settings where self collected samples and urine samples may be more relevant.

3. Nothing is mentioned regarding the financial implications and the time to result for the LFA when the operational characteristics are being evaluated in terms of it meeting the "Target Product Profiles" defined for PoCT for Neisseria gonorrhoeae by the WHO Consultation on PoCT.

4. An important aspect to be re-looked for the LFA would be strengthening the quality assurance of the reader which may give human/ technical errors. (four false negatives were identified within 8 days at a particular site).

5. All these comments become even more significant considering the fact that 80-90% of STI burden exists in developing world where the PoCTS become most relevant due to lack of skilled manpower and infrastructure. These sites may lack the basic facility for sample collection and processing (reader use) making ease-of-use (operational characteristics) most important when using the LFA.

6. PLOS authors have the option to publish the peer review history of their article (what does this mean? ). If published, this will include your full peer review and any attached files.

**Do you want your identity to be public for this peer review?** For information about this choice, including consent withdrawal, please see our Privacy Policy .

Reviewer #1: No

Reviewer #2: **Yes**

---

## [Decision Letter · Decision Letter 1]

12 Dec 2024

Evaluation of a novel point-of-care lateral flow assay screening for Neisseria gonorrhoeae infection among pregnant women in Zimbabwe

PGPH-D-24-01660R1

Dear Dr Kevin Martin,

We are pleased to inform you that your manuscript 'Evaluation of a novel point-of-care lateral flow assay screening for Neisseria gonorrhoeae infection among pregnant women in Zimbabwe' has been provisionally accepted for publication in PLOS Global Public Health.

Best regards,

Gagandeep Singh, M.D.

Academic Editor

Dear Authors,

You have satisfactorily addressed all the queries raised by our two esteemed reviewers.

Reviewer Comments (if any, and for reference):

Reviewer's Responses to Questions

**Comments to the Author**

1. If the authors have adequately addressed your comments raised in a previous round of review and you feel that this manuscript is now acceptable for publication, you may indicate that here to bypass the “Comments to the Author” section, enter your conflict of interest statement in the “Confidential to Editor” section, and submit your "Accept" recommendation.

Reviewer #1: All comments have been addressed

Reviewer #2: All comments have been addressed

2. Does this manuscript meet PLOS Global Public Health’s publication criteria ? Is the manuscript technically sound, and do the data support the conclusions? The manuscript must describe methodologically and ethically rigorous research with conclusions that are appropriately drawn based on the data presented.

Reviewer #1: Yes

Reviewer #2: Yes

3. Has the statistical analysis been performed appropriately and rigorously?

Reviewer #1: Yes

Reviewer #2: Yes

4. Have the authors made all data underlying the findings in their manuscript fully available (please refer to the Data Availability Statement at the start of the manuscript PDF file)?

Reviewer #1: Yes

Reviewer #2: Yes

5. Is the manuscript presented in an intelligible fashion and written in standard English?

Reviewer #1: Yes

Reviewer #2: Yes

6. Review Comments to the Author

Reviewer #1: (No Response)

Reviewer #2: Though most of the comments have been acknowledged, but could not be included in the study at present; addressing the issues in all future studies on the POCT would help you share a more meaningful and comprehensive study.

7. PLOS authors have the option to publish the peer review history of their article (what does this mean? ). If published, this will include your full peer review and any attached files.

**Do you want your identity to be public for this peer review?** For information about this choice, including consent withdrawal, please see our Privacy Policy .

Reviewer #1: No

Reviewer #2: **Yes: ** Dr Aradhana Bhargava
